# Long-Lasting Cognitive Abnormalities after COVID-19

**DOI:** 10.3390/brainsci11020235

**Published:** 2021-02-13

**Authors:** Roberta Ferrucci, Michelangelo Dini, Elisabetta Groppo, Chiara Rosci, Maria Rita Reitano, Francesca Bai, Barbara Poletti, Agostino Brugnera, Vincenzo Silani, Antonella D’Arminio Monforte, Alberto Priori

**Affiliations:** 1Aldo Ravelli Research Center for Neurotechnology and Experimental Brain Therapeutics, University of Milan, 20142 Milan, Italy; roberta.ferrucci@unimi.it (R.F.); michelangelo.dini@gmail.com (M.D.); vincenzo.silani@unimi.it (V.S.); 2ASST-Santi Paolo e Carlo University Hospital, 20142 Milan, Italy; elisabetta.groppo@asst-santipaolocarlo.it (E.G.); chiara.rosci@asst-santipaolocarlo.it (C.R.); mariella.reitano@gmail.com (M.R.R.); francy.bai19@gmail.com (F.B.); antonella.darminio@unimi.it (A.D.M.); 3Department of Health Science (DISS), University of Milan, 20142 Milan, Italy; 4Department of Neurology and Laboratory of Neuroscience, IRCCS Istituto Auxologico Italian, 20149 Milan, Italy; b.poletti@auxologico.it; 5Department of Human and Social sciences, University of Bergamo, 24129 Bergamo, Italy; agostino.brugnera@unibg.it; 6Department of Pathophysiology and Transplantation, “Dino Ferrari” Center, University of Milano, 20122 Milan, Italy

**Keywords:** COVID-19, cognition, processing speed, acute respiratory distress syndrome

## Abstract

Considering the mechanisms capable of causing brain alterations in COVID-19, we aimed to study the occurrence of cognitive abnormalities in the months following hospital discharge. We recruited 38 (aged 22–74 years; 27 males) patients hospitalized for complications of SARS-CoV-2 infection in nonintensive COVID units. Participants underwent neuropsychological testing about 5 months after hospital discharge. Of all patients, 42.1% had processing speed deficits, while 26.3% showed delayed verbal recall deficits. Twenty-one percent presented with deficits in both processing speed and verbal memory. Bivariate analysis revealed a positive correlation between the lowest arterial oxygen partial pressure (PaO_2_) to fractional inspired oxygen (FiO_2_) (P/F) ratio during hospitalization and verbal memory consolidation performance (SRT-LTS score, *r* = 0.404, *p* = 0.027), as well as a positive correlation between SpO_2_ levels upon hospital arrival and delayed verbal recall performance (SRT-D score, *r_s_* = 0.373, *p* = 0.042). Acute respiratory distress syndrome (ARDS) during hospitalization was associated with worse verbal memory performance (ARDS vs. no ARDS: SRT-LTS mean score = 30.63 ± 13.33 vs. 44.50 ± 13.16, *p* = 0.007; SRT-D mean score = 5.95 ± 2.56 vs. 8.10 ± 2.62, *p* = 0.029). Cognitive abnormalities can frequently be found in COVID-19 patients 5 months after hospital discharge. Increased fatigability, deficits of concentration and memory, and overall decreased cognitive speed months after hospital discharge can interfere with work and daily activities.

## 1. Introduction

COVID-19 was initially considered almost exclusively a respiratory syndrome, but increasing evidence indicates that SARS-CoV-2 infection also affects other body districts and functions [1]. More specifically, studies have shown that SARS-CoV-2 is capable of invading the central nervous system (CNS) and causing neurological symptoms [2,3,4,5,6]. Indeed, many coronaviruses are capable of altering the structure and function of the nervous system [7,8]. Additionally, they have been shown to cause nervous system alterations not only through direct infection pathways (both neuronal and circulatory), but also through secondary hypoxia, immune-mediated tissue damage, procoagulative and prothrombotic states, and other mechanisms [9,10].

Neurological symptoms observed in patients with COVID-19 typically include headache, dizziness, myalgia, anosmia, and ageusia [2,4,11,12]. However, more severe complications such as encephalopathy and skeletal muscle injury have also been observed in hospitalized patients [9]. Additionally, growing evidence points towards a notable incidence of cerebrovascular events following SARS-CoV-2 infection [13,14], especially in older patients and more severe cases, but some studies suggest that even younger patients may be at risk [15]. The pathophysiology that underlies cerebrovascular events in patients with COVID-19 is still poorly understood but is likely to be multifactorial. Infection of vascular endothelial cells (which express ACE2 receptors), potential changes of vascular smooth muscle cells (VSMC) in the arteriole, hypercoagulability, and abnormal immune responses can all concur in damaging the vascular system and may increase the risk of cerebrovascular events [16,17,18,19,20,21,22]. Finally, recent evidence indicates that patients who have recovered from COVID-19 might be at increased risk of cognitive decline [23].

Considering the aforementioned mechanisms capable of causing brain alterations in patients with COVID-19, we aimed to study the occurrence of cognitive abnormalities in hospitalized patients in the months after hospital discharge. 

## 2. Materials and Methods

We recruited 38 (aged 22–74; 27 males) patients hospitalized for SARS-CoV-2 infection in various nonintensive COVID units of the ASST Santi Paolo e Carlo hospitals in Milan, Italy, between February and April 2020.

We collected clinical variables such as duration of hospitalization, type and duration of oxygen therapy, viral clearance time (days between first positive and last negative nasopharyngeal swab for SARS-CoV-2), comorbidities, and subjective cognitive deficits. Presence of anosmia/dysgeusia during and/or after hospitalization for COVID-19 was also assessed by asking participants whether they had experienced such symptoms, as initial theories suggested that they could imply viral access to the CNS via retrograde transport through the olfactory pathway [24]. Answers were recorded as binary “yes/no” variables.

We also collected the lowest ratio of arterial oxygen partial pressure (PaO_2_) to fractional inspired oxygen (FiO_2_) (P/F ratio) during the hospital stay, as well as peripheral oxygen saturation (SpO_2_) levels upon hospital arrival. 

Acute respiratory distress syndrome (ARDS) severity was defined by a P/F ratio ≤300 and can be divided into three categories based on the degree of hypoxemia: mild (200 mm Hg < P/F ratio ≤ 300 mm Hg), moderate (100 mm Hg < P/F ratio ≤ 200 mm Hg), and severe (P/F ratio ≤ 100 mm Hg) [25].

Participants underwent the neuropsychological assessment between 4 and 5 months (mean ± SD = 4.43 ± 1.22 months) after hospital discharge. Before proceeding to the full neuropsychological evaluation, patients were screened using the Montreal Cognitive Assessment (MoCA), a screening test for global cognitive functioning [26], in order to exclude those with global cognitive decline or dementia (cutoff > 18.28).

Cognitive functioning was assessed using the Brief Repeatable Battery of Neuropsychological Tests (BRB-NT) [27]. The BRB-NT includes the Selective Reminding Test (SRT), the 10/36 Spatial Recall Test (SPART), the Symbol Digit Modalities Test (SDMT), the Paced Auditory Serial Addition Test (PASAT), and the Word List Generation Test (WLG). The SRT is a test of verbal memory and produces three subscores: (i) SRT-LTS (Long-Term Storage), which reflects the ability to store verbal information in long-term memory; (ii) SRT-CLTR (Consistent Long-Term Retrieval), which reflects the consistency of retrieval from verbal long-term memory storage; and (iii) SRT-D (Delayed Recall), which is a measure of long-term verbal recall ability. The SPART evaluates visuospatial memory and produces two subscores: (i) SPART, a measure of learning and immediate recall, and (ii) SPART-D, a measure of delayed recall. The SDMT is a measure of attention and processing speed, and the score reflects the number of correct symbol-number associations produced by the participant in 90 s. The PASAT evaluates processing speed, working memory, and sustained attention and consists of two tests, one in which numbers are presented with an interval of 3 s (PASAT-3) and one with an interval of 2 s (PASAT-2), the latter being more difficult. Lastly, the WLG is a test of semantic verbal fluency, with the score representing the number of words correctly produced by the participant in 90 s.

Raw scores were adjusted based on published normative data for the Italian version of the BRB-NT. Published normative cutoffs were used to assess the presence of deficits in each BRB-NT subtest (SRT-LTS normative cutoff ≥ 23.3; SRT-CLTR normative cutoff ≥ 15.5; SRT-D normative cutoff ≥ 4.9; SPART normative cutoff ≥ 12.7; SPART-D normative cutoff ≥ 3.6; SDMT normative cutoff ≥ 37.9; PASAT-3 normative cutoff ≥ 28.4; PASAT-2 normative cutoff ≥ 17.1; WLG normative cutoff ≥ 17.0) [27].

We also administered Beck’s Depression Inventory-II (BDI-II) [28], in order to assess whether depressive symptoms negatively impacted cognitive performance [29], and the Subjective Scale of Damage (SSD) questionnaire [30].

Statistical analyses were performed using IBM SPSS 25. Descriptive analyses were performed for demographic and clinical data, as well as for each item of the BRB-NT; normality of distribution was analyzed via the Shapiro–Wilk test. The impact of dichotomous variables (sex, ARDS at hospitalization, presence of hyposmia/dysgeusia, subjective cognitive deficits) on neuropsychological scores was analyzed using Student’s *t*-test for normally distributed variables and Mann–Whitney U test for non-normally distributed variables. Bivariate correlations between continuous variables were analyzed using Pearson’s correlation coefficient (*r*) for normally distributed variables and Spearman’s rank correlation coefficient (*r_s_*) for non-normally distributed variables. Variables that were found to correlate with BRB-NT scores were then entered into a backward elimination model of linear regression (probability of F-to-remove = *p* ≥ 0.05), with the BRB-NT item score as the dependent variable, in order to analyze their predictive value in regards to cognitive status at follow-up.

## 3. Results

### 3.1. Descriptive Analysis

Demographic and clinical data did not differ significantly between males and females (see Table 1 for *t*-test results of mean differences between males and females). Twenty-nine patients (76.3%) received low-intensity oxygen therapy (face mask), while nine (23.7%) did not require oxygen therapy. Of all patients, 55.3% reported the occurrence of either hyposmia or dysgeusia during the course of the illness, 44.7% reported both symptoms, 5.3% reported only dysgeusia, and 5.3% reported only hyposmia. Furthermore, 31.6% reported subjective cognitive decline, and there were cardiovascular comorbidities (e.g., hypertension, diabetes, cardiopathy) in 42%.

Thirty participants completed the SSD questionnaire; of these, 50% reported a moderate to severe increase in fatigability (moderate = 30%, severe = 20%), 26.7% reported a moderate to severe increase in forgetfulness and lack of concentration (moderate = 20%, severe = 6.7%), 23.3% reported a moderate to severe increase in time needed to perform tasks such as reading/writing documents (moderate = 13.3%, severe = 10%), and 20% reported moderate to severe difficulties in learning new skills or procedures (moderate = 13.3%, severe = 6.7%).

A descriptive analysis of neuropsychological scores revealed that 60.5% of our sample had obtained scores below Italian normative cutoffs [27] in at least one task of the BRB-NT. Additionally, 36.8% of patients showed deficits in at least two tasks, 26.3% showed deficits in at least three tasks, and 15.8% showed deficits in four or more tasks.

Of all patients, 42.1% showed processing speed deficits (SDMT score < 37.9), 26.3% showed delayed verbal recall deficits (SRT-D score < 4.9), and 10.5% showed deficits in immediate verbal recall (SRT-LTS score < 23.3; SRT-CLTR score < 15.5). Visual long-term memory (SPART-D score < 3.6) was impaired in 18.4% of patients, and visual short-term memory (SPART score < 12.7) was impaired in 15.8%. PASAT scores below normative cutoffs were obtained by 10.5% (PASAT-3 score) and 5.3% (PASAT-2 score < 17.1) of patients. Semantic verbal fluency deficits (WLG score < 17.0) were observed in 7.9% of patients. Mean scores with standard deviation and normative cutoff values for each test are displayed in Table 2.

A descriptive analysis of BDI-II scores revealed that only 6/38 (15.79%) patients obtained scores above the cutoff (<13), according to Italian normative data [31], indicating the presence of mood disturbances. Of these patients, three reported mild depressive symptoms (BDI-II score 14–19), two reported moderate depressive symptoms (BDI-II score 20–29), and one reported severe depressive symptoms (BDI-II score > 30) [31]. None of these six patients had a documented clinical history of depressive disorders or depressive episodes prior to SARS-CoV-2 infection.

### 3.2. Demographic and Clinical Differences

We did not observe sex-related differences in BRB-NT subtests scores, with the exception of SDMT scores, where females obtained higher scores (SDMT mean scores, females vs. males = 45.35 ± 8.16 vs. 36.94 ± 9.86, *p* = 0.017).

Females more frequently reported a subjective decline in cognitive performance following hospitalization (OR = 7.35, 95% CI 1.53–35.28, *p* = 0.018).

We divided our sample in two groups based on the median age for the total sample (median age = 54) and conducted an independent t-test analysis of differences in BRB-NT scores between the two resulting groups. Participants aged ≥ 55 (n = 20) obtained lower scores in all measures of verbal memory, when compared to those aged < 55 (n = 18) (SRT-LTS mean score: 34.85 ± 13.18 vs. 44.89 ± 13.04, *p* = 0.025; SRT-CLTR mean score: 26.39 ± 10.17 vs. 36.61 ± 15.55, *p* = 0.023; SRT-D mean score: 6.56 ± 2.85 vs. 8.40 ± 2.38, *p* = 0.037). We did not observe statistically significant differences in other BRB-NT subtests.

No statistically significant differences were observed in clinical data (P/F ratio, SpO_2_, duration of hospitalization), cognitive performance (BRB-NT scores), or depression severity (BDI-II scores) between participants who reported the occurrence of hyposmia and/or dysgeusia and those who did not. 

Presence of cardiovascular comorbidities was associated with older age (mean age, 58.19 ± 11.86 vs. 50.00 ± 12.30, *p* = 0.047), but did not determine statistically significant differences in cognitive performance, as measured by BRB-NT scores.

### 3.3. Correlations

Bivariate analysis results (Table 3) revealed a positive correlation between the lowest P/F ratio during hospitalization and verbal memory consolidation performance (SRT-LTS score, *r* = 0.404, *p* = 0.027), while there was no significant correlation between the lowest P/F ratio and other BRB-NT subtests (Figure 1). SpO_2_ levels upon hospital arrival were positively correlated with delayed verbal recall performance (SRT-D score, *r_s_* = 0.373, *p* = 0.042), but not with other BRB-NT subtests. No significant correlation was found between viral clearance time and BRB-NT subtest scores. BDI-II scores correlated negatively with delayed verbal recall performance (SRT-D scores, *r_s_* = −0.372, *p* = 0.023), but not with other BRB-NT subtests.

Multiple linear regression (backward method) was conducted to assess age and the lowest P/F ratio during hospitalization as predictors of verbal memory consolidation performance (SRT-LTS score). Only the lowest P/F ratio during hospitalization remained a predictor of SRT-LTS score (F[1, 28] = 5.449, *p* = 0.027, standardized B = 0.404, Adjusted R^2^ = 0.133).

### 3.4. ARDS vs. No ARDS

Based on the lowest P/F ratios during hospitalization, 21 participants were classified as “no ARDS”, 10 were classified as ‘mild ARDS’, and 2 were classified as “moderate ARDS”. Since most ARDS cases were mild, and only two cases presented moderate severity, we considered the presence of ARDS during hospitalization as a dichotomous yes/no variable. Analyzing differences in demographic and neuropsychological scores based on the presence/absence of ARDS at the time of hospitalization (Table 4), we observed that ARDS at hospitalization was associated with older age (ARDS vs. no ARDS = 60.00 ± 9.64 vs. 49.48 ± 13.74, *p* = 0.027) and worse verbal memory performance, as evidenced by worse verbal long-term memory storage efficiency (SRT-LTS mean score, ARDS vs. no ARDS = 30.63 ± 13.33 vs. 44.50 ± 13.16, *p* = 0.007) and worse delayed verbal recall performance (SRT-D mean score, ARDS vs. no ARDS = 5.95 ± 2.56 vs. 8.10 ± 2.62, *p* = 0.029) (Figure 2).

## 4. Discussion

Five months after hospital discharge, 60.5% of hospitalized COVID-19 patients had cognitive abnormalities: 42% showed a slowing of cognitive processing speed (as evidenced by low SDMT scores) and about 20% showed long-term verbal and spatial memory dysfunctions.

Our data expand previous observations conducted either during hospital stay or at shorter time intervals after hospital discharge. For instance, Helms et al. [32] studied 58 patients with COVID-19 during hospitalization in the ICU and found that 15 of 45 had a dysexecutive syndrome (inattention, disorientation, and difficulties organizing response to command). The study omitted characteristics of the patients who exhibited the dysexecutive syndrome, including age, pre-existing medical conditions, and treatments during the ICU stay. Zhou et al. [33] assessed cognitive function 3 weeks after hospital discharge of 29 patients with COVID-19, reporting a dysfunction in the sustained attention domain and a correlation between serum C-reactive protein (CRP) level and reaction time. Our data add the important information that cognitive abnormalities persist in the months following hospital discharge and may affect non-ICU patients as well.

Additionally, we showed that mental processing speed reduction is not related to clinical characteristics such as SpO_2_ and P/F. We can therefore speculate that it results from brain alterations directly related to viral neurotropism and is not secondary to generalized hypoxemia or other systemic consequences of COVID-19. The presence of SARS-CoV-2 in the brain has been observed by postmortem studies and in vitro studies utilizing brain organoids [34,35,36].

About one-third of acute/subacute patients with COVID-19 referred for neuroimaging show brain abnormalities suggestive of COVID-19-related etiology. The predominant neuroimaging features are diffuse cerebral white matter (WM) hypodensities/hyperintensities attributable to leukoencephalopathy, leukoaraiosis, or rarefied WM [37]. White matter hyperintensities (WMHs) have been extensively associated with cognitive impairment; specifically, the most important WMH-mediating effect was found for processing speed [38].

In our study, patients who had ARDS during hospitalization performed worse on verbal memory tests; this finding is consistent with memory impairment following hospitalization for ARDS [39,40]. Additionally, a meta-analysis of studies on hospitalized patients with SARS and MERS found that 18.9% (95% CI 14.1–24.2) presented memory impairment in the post-illness stage [41]. Prolonged hypoxemia is a cardinal feature of ARDS and can lead to hypoxia-related long-term cognitive impairment [42]. The association between ARDS and verbal memory deficits could be explained by the known sensitivity of medial temporal lobe structures to hypoxic injury [43]. Interestingly, we found significant differences in the delayed recall score (SRT-D) and in the score reflecting memory consolidation efficiency (SRT-LTS) but not in the score reflecting the consistency of stored memory retrieval (SRT-CLTR). This suggests that verbal memory deficits observed in the ARDS group could be associated mainly with impaired memory consolidation, a cognitive process classically related to limbic temporal lobe structures [44]. Consistently, a CT-scan imaging study of 15 patients post-ARDS revealed bilateral temporal horn enlargement [45]. We could therefore hypothesize that memory impairments observed in patients with COVID-19 in the months following hospital discharge are related to hypoxic factors. Aside from hypoxemia, ARDS-mediated neurological damage has also been theorized to involve cytokine-mediated damage following hyperinflammation due to lung injury or sepsis [46,47]. Additionally, mechanical ventilation, hemodynamic instability, blood–brain barrier dysfunction, and hyperinflammation have all been associated with a higher risk of long-term cognitive impairment in patients hospitalized for ARDS [48].

Finally, contrary to what we expected and to what was observed in a survey of the general population [49], the incidence of psychological sequelae was not particularly high, with only 16% (15.79%) of participants reporting clinically relevant depressed mood at the time of neuropsychological testing.

Some key limitations need to be considered when interpreting the results of the present study. Firstly, we were unable to recruit a control sample, which ideally would consist of age-matched patients hospitalized for respiratory problems not related to SARS-CoV-2 infection. Secondly, baseline cognitive scores, which would have been useful for interpreting the cognitive impact of hospitalization for COVID-19, were not available. Additionally, our study lacks direct measures of viral load or inflammatory response, which would have been needed to confirm the results of previous studies linking inflammatory response to subsequent deficits of processing speed. Lastly, our sample consisted predominantly of male subjects, since males tended to be more severely affected by COVID-19 symptoms and required hospitalization more frequently than females; this reduces the generalizability of our results to female patients. Future studies will need to focus on the relation between clinical measures of pulmonary function (e.g., P/F), inflammatory response (e.g., CRP), viral load, and hypercoagulability (e.g., D-dimer) and the risk of developing cognitive deficits following hospitalization for COVID-19.

## 5. Conclusions

Cognitive abnormalities can be frequently found months after hospital discharge in COVID-19 patients. Slowed cognitive processing speed and memory impairment could interfere with patients’ daily functioning and ability to return to work. This latter conclusion is of specific interest for health professional workers, particularly for those whose role requires making quick decisions on a daily basis (e.g., surgeons, first responders, and emergency room personnel). Increased fatigability and deficits of concentration, memory, and overall cognitive speed are reported months after hospital discharge and could interfere with work and daily living. Younger patients and essential workers may therefore benefit from early neuropsychological assessments in order to evaluate the degree of impairment following hospitalization for COVID-19 and its impact on their ability to return to work. Cognitive rehabilitation interventions aimed at enhancing processing speed and memory should also be considered for these populations. Future studies will need to carefully assess the long-term progression of cognitive disturbances in recovered COVID-19 patients, as well as the effectiveness of rehabilitation interventions, particularly on younger patients.

## Figures and Tables

**Figure 1 brainsci-11-00235-f001:**
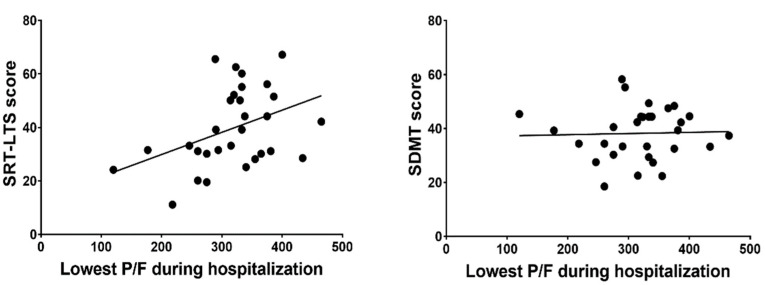
P/F = arterial oxygen partial pressure (PaO_2_)/fractional inspired oxygen (FiO_2_) ratio; SRT-LTS = Serial Recall Test Long-Term Storage; SDMT = Symbol-Digit Modalities Test.

**Figure 2 brainsci-11-00235-f002:**
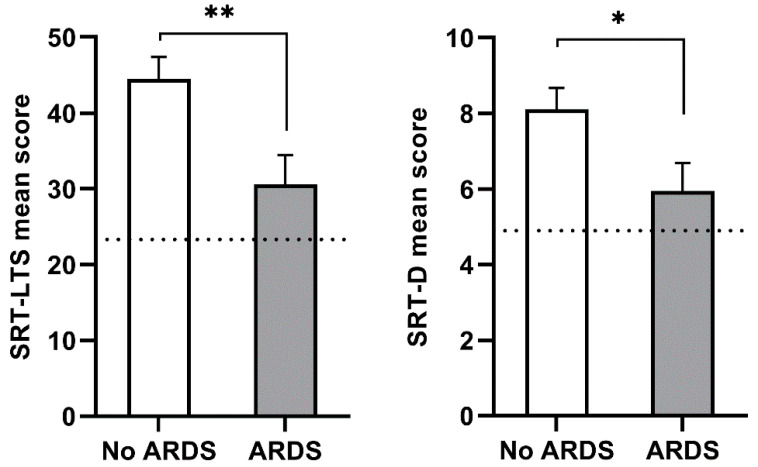
Verbal memory in patients with acute respiratory distress syndrome (ARDS) and without (no ARDS). **Left**: Serial Recall Test Long-Term Storage (SRT-LTS); **right**: Serial Recall Test Delayed Recall (SRT-D). Data are presented as a mean score with standard error. The dotted line indicates normative cutoff scores for the SRT-LTS (cutoff ≥ 23.3) and SRT-D (cutoff ≥ 4.9) subtests. Asterisks denote statistical significance (* = *p* < 0.05; ** = *p* < 0.01).

**Table 1 brainsci-11-00235-t001:** Demographic and clinical data of our sample.

	Females (n = 11)		Males (n = 27)		Total (n = 38)		
	Mean	SD	Mean	SD	Mean	SD	*p*-Value
Age	53.82	12.62	53.30	12.88	53.45	12.64	0.910
Education (years)	11.00	3.71	12.96	2.92	12.39	3.24	0.087
Days of hospitalization	9.55	4.11	9.96	3.95	9.84	3.95	0.772
SpO_2_ upon hospital arrival	96.33	1.97	96.73	2.25	96.62	2.15	0.430
Lowest P/F during hospitalization	326.71	46.44	314.43	78.44	317.30	71.70	0.699
Days between hospital discharge and cognitive assessment	154.18	40.12	123.08	31.08	132.86	36.62	0.016
MoCA adjusted score	25.96	2.20	25.90	2.68	25.92	2.53	0.953

MoCA = Montreal Cognitive Assessment; P/F = arterial oxygen partial pressure (PaO_2_)/fractional inspired oxygen (FiO_2_); SpO_2_ = peripheral oxygen saturation. *p*-values indicate statistical significance of differences between males and females, assessed via independent samples *t*-test.

**Table 2 brainsci-11-00235-t002:** BRB-NT subitem mean scores for the entire sample and normative cutoffs.

	Mean	SD	Normative Cutoff	% Under Normative Cutoff
SRT-LTS score	40.11	13.88	≥23.3	10.5%
SRT-CLTR score	31.77	14.09	≥15.5	10.5%
SPART score	17.63	5.08	≥12.7	15.8%
SDMT score	39.37	10.07	≥37.9	42.1%
PASAT-3 score	43.39	10.64	≥28.4	10.5%
PASAT-2 score	32.53	9.56	≥17.1	5.3%
SRT-D score	7.53	2.74	≥4.9	26.3%
SPART-D score	5.76	1.91	≥3.6	18.4%
WLG score	25.65	5.23	≥17.0	7.9%

SRT-LTS = Serial Recall Test Long-Term Storage; SRT-CLTR = Serial Recall Test Consistent Long-Term Retrieval; SRT-D = Serial Recall Test (Delayed Recall); SPART = Spatial Recall Test; SPART-D = Spatial Recall Test (Delayed Recall); SDMT = Symbol-Digit Modalities Test; PASAT = Paced Auditory Serial Addition Test; WLG = Word List Generation.

**Table 3 brainsci-11-00235-t003:** Correlations between neuropsychological test scores and clinical data.

		Lowest P/F	SpO_2_ upon Hospital Arrival	Days of Hospitalization	BDI-II
SRT-LTS	Correlation coefficient	**0.404**	0.240	−0.206	−0.160
	*p*	**0.027**	0.201	0.222	0.344
SRT-CLTR	Correlation coefficient	0.241	0.230	−0.108	−0.186
	*p*	0.199	0.221	0.524	0.270
SRT-D	Correlation coefficient	0.318	**0.373**	−0.020	**−0.372**
	*p*	0.087	**0.042**	0.906	**0.023**
SPART	Correlation coefficient	−0.044	−0.007	−0.063	0.194
	*p*	0.817	0.971	0.713	0.250
SPART-D	Correlation coefficient	0.080	−0.013	−0.088	0.064
	*p*	0.674	0.944	0.604	0.709
SDMT	Correlation coefficient	0.032	−0.024	0.028	0.139
	*p*	0.866	0.902	0.868	0.414
PASAT-3	Correlation coefficient	0.163	0.158	−0.242	0.193
	*p*	0.389	0.406	0.149	0.254
PASAT-2	Correlation coefficient	0.216	0.134	−0.141	0.009
	*p*	0.251	0.479	0.404	0.960
WLG	Correlation coefficient	0.179	0.328	0.194	−0.252
	*p*	0.345	0.077	0.250	0.133

In bold: statistically significant correlations (*p* < 0.05). P/F = arterial oxygen partial pressure (PaO_2_)/fractional inspired oxygen (FiO_2_); SpO_2_ = peripheral oxygen saturation; BDI-II = Beck’s Depression Inventory -II; SRT-LTS = Serial Recall Test Long-Term Storage; SRT-CLTR = Serial Recall Test Consistent Long-Term Retrieval; SRT-D = Serial Recall Test (Delayed recall); SPART = Spatial Recall Test; SPART-D = Spatial Recall Test (Delayed recall); SDMT = Symbol-Digit Modalities Test; PASAT = Paced Auditory Serial Addition Test; WLG = Word List Generation.

**Table 4 brainsci-11-00235-t004:** Neuropsychological scores in patients with Acute Respiratory Distress Syndrome (ARDS) and patients without it (No ARDS).

	No ARDS		ARDS		
	Mean	SD	Mean	SD	*p*-Value
SRT-LTS	44.50	13.16	30.63	13.33	**0.007**
SRT-CLTR	34.42	14.46	25.59	14.68	0.103
SRT-D	8.10	2.62	5.95	2.56	**0.029**
SPART	17.49	4.89	17.49	4.87	0.998
SPART-D	5.73	1.86	5.30	1.89	0.526
SDMT	37.15	8.57	38.73	11.49	0.658
PASAT-3	43.70	1.78	41.13	9.89	0.503
PASAT-2	33.52	1.23	3.20	8.80	0.355
WLG	26.99	4.47	23.62	5.84	0.073

Data are displayed as mean and standard deviation (SD); in bold: statistically significant differences (*p* < 0.05). SRT-LTS = Serial Recall Test Long-Term Storage; SRT-CLTR = Serial Recall Test Consistent Long-Term Retrieval; SRT-D = Serial Recall Test Delayed recall; SPART = Spatial Recall Test; SPART-D = Spatial Recall Test Delayed recall; SDMT = Symbol-Digit Modalities Test; PASAT = Paced Auditory Serial Addition Test; WLG = Word List Generation.

## Data Availability

Data not provided in the article because of space limitations will be shared at the request of other investigators for purposes of replicating procedures and results.

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
