# Peer review of "Long-Lasting Cognitive Abnormalities after COVID-19"

_brainsci, 2021, doi:10.3390/brainsci11020235_

Round 1

Reviewer 1 Report

The manuscript by Ferrucci et al., was a relative comprehensive research article on the potential effects of clinical management on the psychological outcomes of COVID-19 patients. The authors focused on the potential cognition and subjective cognitive decline and correlated with SARS-CoV-2 infection. The investigation was well performed and data was well collected. On the other hand, the study seemed to involve aging issue, which might be more suitable for more specific journal. There were some moderate concerns:

  • Lines 48-51, missing vascular cellular mechanisms, such as potential changes of VSMC in the arteriole.
  • Lines 57-60, it was quite understood that the study was primarily designed for non-ICU patients and that the viral clearance during/in-between the tests were obtained. Since COVID-19 mechanisms were still unknown, it needed the viral load information. To improve the quality of the research, the data was needed.
  • It was unknown why the 5 months time point was selected for neuropsychological assessment.
  • Section Results3.1, Lines 116-118, “did not differ significantly between male and female” missing statistics. In addition, Table 2 needs analysis on male and female reported too.
  • Line 165 “participants aged”at n=20 among all groups. Missing control aged group.  
  • Rationale for hyposmia and dysgeusia in the analysis was unclear.
  • Depression was involved in the analysis, but it was unknown why DSM-5 was not used.
  • Figure 1 missing viral load information.
  • Figure 2 missing control group (no symptoms or normal people).
  • In Discussion section, Lines 221-222, “decrease of mental processing speed” and “memory dysfunctions” were unclear.

Author Response

We thank you for the valuable comments and we have modified the manuscript accordingly so that we feel it substantially improved. We thank this reviewer for their comments, and hope to have addressed their concerns appropriately.

1. “Lines 48-51, missing vascular cellular mechanisms, such as potential changes of VSMC in the arteriole.”

Reply: as suggest by the reviewer we added vascular muscular cells infection as a potential mechanism of vascular damage. References have been updated accordingly. 

2. “Lines 57-60, it was quite understood that the study was primarily designed for non-ICU patients and that the viral clearance during/in-between the tests were obtained. Since COVID-19 mechanisms were still unknown, it needed the viral load information. To improve the quality of the research, the data was needed.”

Reply: This is an important concern, but data on viral load were sadly unavailable to us at the time of data collection; we have added a section at the end of the discussion, detailing this and other limitations.

3. “It was unknown why the 5 months time point was selected for neuropsychological assessment.”

Reply: Patients were assessed at approximately 4-5 months after hospital discharge due to some key reasons: 1) We aimed to assess lasting cognitive alterations in recovered COVID-19 patients, as opposed to cognitive deficits observable during hospitalization, or in the weeks immediately after hospital discharge; 2) Lockdown measures in Italy began progressively relaxing in June 2020, and recruiting previously hospitalized patients for face-to-face assessments for research purposes was considered safe.

4. “Section Results3.1, Lines 116-118, “did not differ significantly between male and female” missing statistics. In addition, Table 2 needs analysis on male and female reported too.”

Reply: We have added statistical significance values for differences between males and females in Table 1

5. “Line 165 “participants aged”at n=20 among all groups. Missing control aged group. “

Reply: The design of our study lacked a control group (normal people); we have added a section at the end of the discussion, detailing this and other limitations.

6. “Rationale for hyposmia and dysgeusia in the analysis was unclear.”

Reply: We have added a brief rationale for the inclusion of hyposmia/dysgeusia as a clinical variable

7. “Depression was involved in the analysis, but it was unknown why DSM-5 was not used.”

Reply: The reviewer is right, but a clinical interview required for the diagnosis of depressive disorders was not conducted in our study. Results have been reworded; it should now be clearer that they reflect self-reported depressive symptoms, and not a clinical diagnosis of depression.

8. “Figure 1 missing viral load information.”

Reply: This information was sadly unavailable to us at the time of data collection; we have added a section at the end of the discussion, detailing this and other limitations.

9. “Figure 2 missing control group (no symptoms or normal people).”

Reply: The design of our study lacked a control group (normal people); we have added a section at the end of the discussion, detailing this and other limitations.

10. “In Discussion section, Lines 221-222, “decrease of mental processing speed” and “memory dysfunctions” were unclear.”

Reply: We have provided better clarification regarding observed processing speed deficits and memory dysfunctions, which should render these lines clearer.

Reviewer 2 Report

Good job on your paper that looks at cognitive abnormalities after 4-5 months of hospitalization. 

The majority of your sample was male (27 of 38 patients) who underwent neuropsychological testing at 123.08 days (about 4 months). Why do you report that neuropsychological testing was conducted at about 5 months? It would be preferable to report the range.

You report that pts underwent neuropsychological testing around 5 months after hospital discharge. It would be better to explain it in terms of the time range. For eg. Something like 4.1-5.1 months after discharge with a mean of 4.3.

The following papers may be helpful to include as additional references:

1) Immediate and long-term consequences of COVID-19 infections for the development of neurological disease. (Heneka et al)

2) The ‘third wave’: impending cognitive and functional decline in COVID-19 survivors. (Baker et al)

3) Cognitive and Neuropsychiatric Manifestations of COVID-19 and Effects on Elderly Individuals With Dementia (Alonso-Lan et al)

You may expand the discussion and include more information on the limitations of your study including the limitations of neuropsychological testing. 

Author Response

We thank this reviewer for their comments, and hope to have addressed their concerns appropriately.

1. “The majority of your sample was male (27 of 38 patients) who underwent neuropsychological testing at 123.08 days (about 4 months). Why do you report that neuropsychological testing was conducted at about 5 months? It would be preferable to report the range. You report that pts underwent neuropsychological testing around 5 months after hospital discharge. It would be better to explain it in terms of the time range. For eg. Something like 4.1-5.1 months after discharge with a mean of 4.3.”

Reply: We agree with your suggestion; we have amended the section accordingly

2. “The following papers may be helpful to include as additional references:

1) Immediate and long-term consequences of COVID-19 infections for the development of neurological disease. (Heneka et al)
2) The ‘third wave’: impending cognitive and functional decline in COVID-19 survivors. (Baker et al)
3) Cognitive and Neuropsychiatric Manifestations of COVID-19 and Effects on Elderly Individuals With Dementia (Alonso-Lan et al)”

Reply: As suggest by the reviewer we have added the suggested references in relevant sections of the introduction

3. “You may expand the discussion and include more information on the limitations of your study including the limitations of neuropsychological testing. “

Reply: The reviewer is right. We have now improved the paragraph discussing possible limitations

Reviewer 3 Report

I read this report with interest and feel that these data are well-conceptualized and presented. The writing is clear, and the statistical analyses are sound.  Choice of neuropsychological test battery is sound.  Mere reporting on the types of cognitive abnormalities found in post-hospitalized covid 19 patients is very useful.

I have a few comments that I believe would enhance this report.

First, would be interested in the number of patients that were excluded because of global cognitive deficits detected by the MOCA.  It seems that this is an important group and should be mentioned in the manuscript. Is it a high or low number?  Also, is the MOCA a useful screening or bedside measure to detect deficits in processing speed, executive function or verbal memory?. The average MOCA score was about 26, so I suspect that there were some participants who scored lower on this measure, and some higher. There might be room for correlation.

The authors mention the issue of elevation in inflammatory cytokine levels as a potential cause of neurocognitive changes during and over time in COVID-19, however, they do not include markers of inflammation in their study. I wonder if, in addition to measures of oxygen deprivation/depletion, there were inflammatory markers (e.g., CRP, ferritin, ESR, etc.) drawn during the course of the hospitalization, that could be correlated with subsequent cognitive measures. Measurement of inflammatory markers concurrent with NP testing could be revealing as well. In either case, a measure of "inflammatory load" would be interesting.  This might further strengthen their contention in the discussion that viral neurotropism explains reductions in processing speed.  If the authors do not have such data, this should be discussed in the limitations and suggested for future research.  

I would ask a similar question about d-dimer, which is also routinely measured in clinical care of COVID-19 patients, and reflects hypercoagulability.

Finally, in the last paragraph, it would be helpful to further flesh out the functional implications of the findings. Rather than say function is affected, how is it affected, i.e. would individuals have a more difficult time making rapid “emergency” decisions, operating machinery, etc. and what type of clinical advice or remediation could be offered or research would be recommended?

Author Response

We thank the reviewer for their comments. Here is a summary of amendments that were carried out following your suggestions.

1. First, would be interested in the number of patients that were excluded because of global cognitive deficits detected by the MOCA.  It seems that this is an important group and should be mentioned in the manuscript. Is it a high or low number? 

Reply: The reviewer is right, this is an important concern, but in our study no participants were excluded based on MoCA score, as the minimum observed adjusted score (19.52) was greater than the adopted cut-off score. This is due, in our opinion, to the fact that the majority of subjects were aged <65, which decreases the likelihood of observing significant global cognitive decline.

2. Also, is the MOCA a useful screening or bedside measure to detect deficits in processing speed, executive function or verbal memory? The average MOCA score was about 26, so I suspect that there were some participants who scored lower on this measure, and some higher. There might be room for correlation.

Reply: We selected the MoCA only as a screening tool for global cognitive decline, several studies support the use of the MoCA as a screening tool for the presence of cognitive dysfunction characterized by deficits of processing speed and memory. We observed that 94.6% of subjects scored above the 50th percentile, according to published Italian normative values, we found that only 2 subjects obtained scores 1.5 SD below the observed mean (scores <22,2) , and 3 subjects scored 1.5 SD above the mean (scores >29,8).

(Santangelo et al. Normative data for the Montreal Cognitive Assessment in an Italian population sample. Neurol Sci. 2015 Apr;36(4):585-91. doi: 10.1007/s10072-014-1995-y)

3. The authors mention the issue of elevation in inflammatory cytokine levels as a potential cause of neurocognitive changes during and over time in COVID-19, however, they do not include markers of inflammation in their study. I wonder if, in addition to measures of oxygen deprivation/depletion, there were inflammatory markers (e.g., CRP, ferritin, ESR, etc.) drawn during the course of the hospitalization, that could be correlated with subsequent cognitive measures. Measurement of inflammatory markers concurrent with NP testing could be revealing as well. In either case, a measure of "inflammatory load" would be interesting.  This might further strengthen their contention in the discussion that viral neurotropism explains reductions in processing speed.  If the authors do not have such data, this should be discussed in the limitations and suggested for future research.  I would ask a similar question about d-dimer, which is also routinely measured in clinical care of COVID-19 patients, and reflects hypercoagulability.

Reply: The reviewer is right. We have now improved the paragraph discussing possible limitations

4. Finally, in the last paragraph, it would be helpful to further flesh out the functional implications of the findings. Rather than say function is affected, how is it affected, i.e. would individuals have a more difficult time making rapid “emergency” decisions, operating machinery, etc. and what type of clinical advice or remediation could be offered or research would be recommended?

Reply: We have expanded upon our conclusions, as per your suggestion

Round 2

Reviewer 1 Report

The manuscript entitled by “Long-lasting cognitive abnormalities after COVID-19” was a relatively comprehensive research article on the potential effects of SARS-CoV-2 infection on the behavior outcome in patients with COVID-19. The authors focused on cognitive and depressive changes long-term following the recovery. Investigation was well performed. There were some moderate concerns:

  • Page 2 Lines 48-50 “smooth muscle cells (which express ACE2 receptors)”were proposed by the authors; however it seemed unknown whether VSMC can express ACE2 which leads to viral infection. References 16 and 17 were not directly relevant. The authors need to provide more evidence/references.
  • Page 3 Lines 145-147, add normal control ratios for presence of mood disturbances.
  • Page 2 Line 63 and , “anosmia”and “dysgeusia” assessment was not described. The olfactory pathway was mentioned (as in Line 65), but the olfactory functions and related analysis were missing.

Author Response

1. Page 2 Lines 48-50 “smooth muscle cells (which express ACE2 receptors)”were proposed by the authors; however it seemed unknown whether VSMC can express ACE2 which leads to viral infection. References 16 and 17 were not directly relevant. The authors need to provide more evidence/references.

Reply: As suggest by the reviewer we have added more references which are more relevant and specific in relation to what is being discussed in the paragraph.

2. Page 3 Lines 145-147, add normal control ratios for presence of mood disturbances.

Reply: In our study, we administered Beck’s Depression Inventory–II (BDI-II), in order to evaluate the presence of mood disturbance that could have impacted negatively on cognitive performance. We used Italian normative data for the BDI-II (page 3, line 149; page 4,   line 152), which are based on a sample of healthy subjects.

3. Page 2 Line 63 and , “anosmia” and “dysgeusia” assessment was not described. The olfactory pathway was mentioned (as in Line 65), but the olfactory functions and related analysis were missing.

Reply: The reviewer is right. We have now improved the paragraph providing a more detailed account of the assessment of anosmia and dysgeusia (page 2, lines 63-66). Presence of alterations of smell and/or taste were assessed by asking participants whether they had experienced hyposmia and/or dysgeusia, and answers were recorded as binary ‘yes/no’ variables.
We expanded upon the ‘descriptive analysis’ of hyposmia and dysgeusia (page 3, lines 123-125). 55.3% of patients reported the occurrence of either hyposmia or dysgeusia during the course of the illness, 44.7% of patients reported both symptoms, 5.3% reported only dysgeusia, and 5.3% reported only hyposmia. Presence of hyposmia and/or dysgeusia was assessed qualitatively, with the intent of evaluating whether the occurrence of these symptoms was associated with different cognitive outcomes.